# Culture Conditions Affect Antioxidant Production, Metabolism and Related Biomarkers of the Microalgae *Phaeodactylum tricornutum*

**DOI:** 10.3390/antiox11020411

**Published:** 2022-02-17

**Authors:** Eleonora Curcuraci, Simona Manuguerra, Concetta Maria Messina, Rosaria Arena, Giuseppe Renda, Theodora Ioannou, Vito Amato, Claire Hellio, Francisco J. Barba, Andrea Santulli

**Affiliations:** 1Department of Earth and Marine Sciences DiSTeM, University of Palermo, Via Barlotta 4, 91100 Trapani, Italy; eleonora.curcuraci@unipa.it (E.C.); simona.manuguerra@unipa.it (S.M.); rosaria.arena@unipa.it (R.A.); giuseppe.renda02@unipa.it (G.R.); andrea.santulli@unipa.it (A.S.); 2Department of Chemistry, Faculty of Science, Aristotle University of Thessaloniki, 541 24 Thessaloniki, Greece; theodora_ioannou1878@yahoo.gr; 3L’Avannotteria Società Agricola a Responsabilità Limitata, Contrada Triglia Scaletta, 91020 Petrosino, Italy; vito.amato@avannotteria.com; 4LEMAR, IRD, CNRS, Ifremer, Université de Brest, F-29280 Plouzane, France; claire.hellio@univ-brest.fr; 5Nutrition and Food Science Area, Preventive Medicine and Public Health, Food Science, Toxicology and Forensic Medicine Department, Faculty of Pharmacy, Universitat de València, Av. Vicent Andrés Estellés, s/n, 46100 Burjassot, València, Spain; 6Istituto di Biologia Marina, Consorzio Universitario della Provincia di Trapani, Via G. Barlotta 4, 91100 Trapani, Italy

**Keywords:** *Phaeodactylum tricornutum*, nitrogen stress, gene expression, lipid biosynthesis, photosynthesis, antioxidant activity

## Abstract

*Phaeodactylum tricornutum* (Bacillariophyta) is a worldwide-distributed diatom with the ability to adapt and survive in different environmental habitats and nutrient-limited conditions. In this research, we investigated the growth performance, the total lipids productivity, the major categories of fatty acids, and the antioxidant content in *P. tricornutum* subjected for 15 days to nitrogen deprivation (N−) compared to standard culture conditions (N+). Furthermore, genes and pathways related to lipid biosynthesis (i.e., glucose-6-phosphate dehydrogenase, acetyl-coenzyme A carboxylase, citrate synthase, and isocitrate dehydrogenase) and photosynthetic activity (i.e., ribulose-1,5-bisphospate carboxylase/oxygenase and fucoxanthin-chlorophyll a/c binding protein B) were investigated through molecular approaches. *P. tricornutum* grown under starvation condition (N−) increased lipids production (42.5 ± 0.19 g/100 g) and decreased secondary metabolites productivity (phenolic content: 3.071 ± 0.17 mg GAE g^−1^; carotenoids: 0.35 ± 0.01 mg g^−1^) when compared to standard culture conditions (N+). Moreover, N deprivation led to an increase in the expression of genes involved in fatty acid biosynthesis and a decrease in genes related to photosynthesis. These results could be used as indicators of nitrogen limitation for environmental or industrial monitoring of *P. tricornutum*.

## 1. Introduction

Microalgae are an exceptionally diverse group of photosynthetic marine microorganisms that represent the most numerous forms of life in marine ecosystems. It is known from the literature that the estimated 72 k species of microalgae provide an important contribution to the marine environment and ecosystems [1,2,3,4].

These microorganisms can grow in a variety of environments and conditions, from the deepest marine system to the polar, playing an essential role in the evolution of life as the planet’s primary producers of oxygen and the most abundant food source for marine animals [2,3,5]. Diatoms are important marine microalgae, interesting not only for their ecological role but also for their ability to synthesize bioactive compounds useful for both human nutrition and nutraceutical and pharmaceutical applications [4,6,7,8,9,10]. In recent decades, the development of molecular biology and the evolution of “omics” techniques have increased knowledge of marine microorganisms, generating considerable academic and industrial interests [4].

Microalgae are largely used in aquaculture as a food source or as wastewater treatment [5], in biofuel production (limiting nitrogen sources to increase their lipid content) [11], and in the cosmeceutical industry for the production of natural bioactive compounds beneficial to humans health [12]. Cellular biochemical pathways of energy transfer and production of bioactive compounds and metabolic switches in microalgae species are still unexplored [13]. For instance, microalgae are able to alter their metabolism and physiology to survive in unfavorable conditions [11]. Numerous studies have shown that microalgae under stress conditions increase lipid production [14,15,16] and decrease photosynthetic pigments as reported by Longworth et al. [17] in *Phaeodactylum tricornutum*. Under optimal conditions, microalgae produces antioxidant bioactive compounds (i.e., phenolic compounds and carotenoids) and fatty acids representing a rich source of PUFAs [18].

*P. tricornutum* exhibits commercial and industrial potentialities, and it is considered a model organism for its adaptable capacity to unfavorable conditions [19] such as nitrogen deprivation. It is widely used for human nutrition thanks to its high-quality protein content and other compounds such as sterols, ω-3 fatty acids useful for human health [20]. Nitrogen is an essential nutrient required by microalgae for their metabolic activities [21]. The reduction of carbon metabolism and the increase in lipid content in microalgae can be associated to a nitrogen deprivation condition [22].

For this reason, the aim of this work was to assess lipid and antioxidant productivity of *P. tricornutum.* The metabolic switches occurring in *P. tricornutum* cultivated under standard and starvation conditions were investigated through molecular markers related to lipid biosynthesis and photosynthesis. The effect of nitrogen stress in *P. tricornutum* was examined at the molecular level by genes related to lipid biosynthesis and photosynthesis: glucose-6-phosphate dehydrogenase (*G6PDH*), acetyl-coenzyme A carboxylase (*ACCase*), citrate synthase (*Cit syn*), isocitrate dehydrogenase (*isocit DH*), ribulose-1,5-bisphospate carboxylase/oxygenase (*RBCL*), and fucoxanthin-chlorophyll a/c binding protein B (*FCP B*) (Figure 1).

## 2. Materials and Methods

### 2.1. Cultivation Conditions

*P. tricornutum* strain AC171 was obtained from the Algobank culture collection (France). Growth experiments for each different nutrient conditions, standard (N+) and starvation (N−), were performed in two replicates in 1.5 L flasks, starting from an initial cell density of 1 million cells/mL (T0). The standard culture medium was prepared according to a modified process of Fanesi et al. [23]: 3.3% *w*/*v* sea salt containing macro-elements and trace elements, 0.054% *v/v* NaNO_3_ (1M), 0.021 % *v/v* Na_2_HPO_4_ (0.1M), 0.01% *v/v* vitamin stock solution (i.e., 297 nM thiamine–HCl, 4.09 nM biotin, and 1.47 nM B_12_), 0.1% trace metal stock solution I (i.e., 6.56 µM FeCl_3_ and 6.56 µM Na_2_EDTA), and trace metal stock solution II (i.e., 2.42 µM MnSO_4_, 8.29 µM Na_2_EDTA, 254 nM ZnSO_4_, 5.69 nM CoSO_4_, 6.10 nM Na_2_MoO_4_, 1.00 nM Na_2_SeO_3_, and 6.30 nM NiCl_2_), 0.2% *v/v* Na_2_SiO_3_ (0.105 M), and 1% *v/v* Tris-HCl (1M) for the maintenance of pH at 8. The starvation culture medium contained the same nutrients, at the same concentrations, as the standard culture, apart from the nitrogen source: NaNO_3_–, which was only 0.0001% *v/v* of the total volume. The medium was autoclaved at 120 °C for 20 minutes before inoculation. The aeration of the cultures, which is essential for the growth and mixing of the cells, was carried out using an aquarium air pump. The airflow was supplied to cultures passing through microfilters (pore size ø = 0.2 μm). Microalgae cultures were kept in aseptic environments to avoid possible contamination at constant temperatures (20 ± 2 °C) and continuous lighting (neon light, 36 W). The cell density was monitored every 3 days using a bright-line hemocytometer (Sigma–Aldrich, Saint Louis, MO, USA ) and an optical microscope from LW Scientific, until reaching the plateau phase. Analyses were conducted on the 15th day of culture before reaching the plateau phase.

### 2.2. Biomass Isolation

For the biomass isolation, microalgae cultures were pelleted by centrifugation at 5000 rpm at 4 °C for 20 min (Eppendorf Centrifuge 5430 R). The pellets were recovered, frozen at −80 °C, and lyophilized (Labconco FreeZone25 freeze dryer, equipped with an Edwards vacuum pump oil model R.V 8) for further analysis.

### 2.3. Lipid Content

#### Total Lipids and Fatty Acids

Total lipids were determined according to Folch’s method [24], quantified gravimetrically, and resuspended in n-hexane. All samples were analyzed in triplicate (*n* = 3). The quantitative determination of fatty acids was conducted from the total lipid content of dried microalgae, according to the method described by Lepage and Roy [25]. Total fatty acids were analyzed as described by Messina et al. [26]. To evaluate the extent of oxidation, the polyene index [27] was determined based on Formula (1):(1)$$PI=\frac{\left(EPA+DHA\right)}{16:0}$$

### 2.4. Antioxidant Assays

Extracts from algal biomass were obtained following the protocol previously described by Safafar et al. [28]. Fifty milligrams of freeze-dried biomass, homogenized with 5 mL of 80% ethanol using an Ultraturrax (T25 basic, Ika), were extracted. Then, the extracts were centrifuged at 5000 rpm at 4 °C for 10 minutes, and the supernatant was separated. After, the extracts were evaluated at different concentrations (8–0.25 mg mL^−1^) for determining antioxidant content. In this study, we performed four assays for the evaluation of the antioxidant activity of microalgae cells.

Polyphenol content was determined spectrophotometrically according to Folin–Ciocalteu [29]. A gallic acid standard solution was prepared in ethanol at concentrations 5–100 mg mL^−1^. The analysis was performed on a 96-well plate, and all samples were analyzed in triplicate. The absorbance was measured at 725 nm, using a spectrophotometer (Multiskan-Sky Microplate Reader, Thermo-Scientific^TM^, Waltham, MA, USA). Total phenolics contents are expressed as a mg of gallic acid mg^−1^ of microalgae weight.

Total carotenoid content was performed spectrophotometrically according to Maadane et al. [30]. Extracts were prepared at concentration of 1 mg mL^−1^ in ethanol. The samples’ absorbances were measured at 470, 648, and 664 nm by spectrophotometer (Multiskan Thermo Fisher Scientific, Waltham, MA, USA). The total carotenoids content was calculated using Equations (2) of Lichtenthaler et al. [31]:(2)$$Chl_{a}=13.36\;x\;Abs_{664}-5.19\;Abs_{648}\;Chl_{b}=27.43\;x\;Abs_{648}-8.12\;Abs_{664}\;\mathrm{Total}\;\mathrm{carotenoids}=\frac{\left(100\;x\;Abs_{470}-1.63\;x\;Chl_{a}-109.96\;x\;Chl_{b}\right)}{221}$$

For the determination of total antioxidant capacity different assays were carried out. The DPPH radical scavenging activity was performed on modified Bernatoniene et al. [32]. Standard curves were prepared for gallic acid using different concentrations (0.1–0.005 mg mL^−1^). For this, 40 μL of extracts were mixed with 0.1 mM DPPH radical solution in ethanol (prepared before the analysis) in a 96-well plate. Absorbance was read after 30 min (Multiskan Thermo Fisher Scientific, Waltham, MA, USA). The percentage of DPPH inhibition was obtained by the following Equation (3):(3)$$\mathrm{Scavenging}\;\mathrm{effect}\;(\%)=\left[1-\left(\frac{absorbance\;of\;sample-absorbance\;of\;blank}{absorbance\;of\;control}\right)\right]*100$$

IC 50, which is the concentration of antioxidants reduced at 50%, was calculated through linear regression analysis.

Cellular antioxidant properties by the reducing power assay were determined according to Falleh et al. [33]. After 20 min incubation at 50 °C, the absorbance was read at 700 nm. EC 50 represents the concentration of extracts in which the absorbance is at the half value, and it was used to assess the reducing power of the samples.

### 2.5. Real-Time PCR Analysis

Total RNA was extracted from fresh samples at time zero and after 15 days of monitoring in nitrogen standard and stress cultures. RNA was isolated from samples in PureZOL™ using the Aurum Total RNA Fatty and Fibrous Tissue Kit (Bio-Rad, Hercules, CA, USA) and was measured spectrophotometrically using a Thermo Scientific™ µDrop Plate Spectrophotometer. All samples were analyzed in triplicate. Then, reverse transcription was performed using the 5X iScript Reaction Mix Kit (Bio-Rad, Hercules, CA, USA) according to manufacturer’s instructions. 

The amplification and the relative quantification were performed in triplicate on genes *G6PDH, rbcL, FCP B*, *ACCase, Cit syn*, and *isocit DH* (Table 1). Relative gene expression was evaluated after normalization with the reference genes. Data processing and statistical analysis were performed using CFX Manager Software (Bio-Rad, Hercules, CA, USA). The relative expression of all genes was calculated by the 2^−ΔΔCT^ method [34] using *P. tricornutum* RPS (ribosomal protein S1) and TBP (TATA box-binding protein) as the endogenous reference.

### 2.6. Statistical Analysis

Statistical analysis was performed using the computer application SPSS for Windows^®^ (version 20.0, SPSS Inc., Chicago, IL, USA). All the analyses were carried out in triplicate. The results are expressed as the mean ± standard deviation. The homogeneity of variance was confirmed by the Levene test. Data were subjected to one-way analysis of variance (ANOVA), and Student–Newman–Keuls or Games–Howell post hoc tests were performed in order to make multiple comparisons between experimental groups. The significance level was 95% in all cases (*p* < 0.05).

## 3. Results and Discussion

### 3.1. Cell Growth

The obtained results in Figure 2 show a reduction in the growth performance of *Phaeodactylum tricornutum* cultured under starvation (N−) conditions as a common response to nutrient limitations that trigger modifications in primary metabolism [39]. A growth rate exponential phase occurred until day 12 in both treatments (Figure 1) even if, in the first 9 days of culture, a faster growth rate in *P. tricornutum* N+ was observed.

From day 3, a colorimetric change in the cultures was evident. The standard cultures (N+) retained a brown color, while in N−, they turned to yellow due to the degradation of pigments. On day 12, N− cultures reached their maximum growth rate, followed by a decrease from day 13 to 15, while N+ cultures continued their exponential phase until the end of monitoring (day 18) (Figure 2).

The yields of N− and N+ were, respectively, 0.1574 ± 0.010 g/L and 0.3026 ± 0.0141 g/L. According to Yodsuwan et al. [14], long-term nutrient stress conditions reduced growth performance in *P. tricornutum*. The stress condition resulting from nutrient deficiency led to a decrease in biomass productivity in favor of the production of bioactive microalgal secondary metabolites [40]. No universal or efficient nutrient acquisition strategy of nitrogen limitation exists to date that increases the amount of bioactive secondary metabolites without affecting algal biomass [41,42]. Mixotrophic cultivation of *P. tricornutum* [43] and *Chlorella vulgaris* (Chlorophyta) [44], in long-term nitrogen-limited cultivations, resulted in improved biomass and biomolecule productivity. Integrative approaches include the synergistic action of different parameters that control both the growth rate and production of bioactive compounds such as light, temperature, and pH [45]. Burch and Franz [46] suggested a combination of nitrogen starvation and low concentration of H_2_O_2_ as a chemical modulator to stimulate triacylglycerol (TAG) synthesis in the early exponential phase without changing biomass productivity.

### 3.2. Lipid Content

#### 3.2.1. Total Lipid Content

Manipulation of microalgae culture medium can induce a variation in lipid concentrations, especially nitrogen limitation conditions lead to an increase in lipid contents [39]. Throughout *P. tricornutum* monitoring for 15 days in N−, the quantity of total lipids increased compared to the initial culture condition (T0) and to the lipid content in N+, as shown in Figure 3, reaching 42.5 ± 0.19 g/100 g of dried algal biomass in N− and 33.35 ± 0.12 g/100 g of dried algal biomass in N+. Our results are in accordance with studies that focused on the cultivation of *P. tricornutum* under nutrient stress conditions [47] on *Nannochloropsis* sp. (Ochrophyta, Eustigmatophyceae) [48], *Conticribra weissflogii* (formerly *Thalassiosira weissflogii*), and *Cyclotella cryptica* (Bacillariophyta) [49] maintained under nitrogen deprivation.

#### 3.2.2. Fatty Acid Content

The ability of microalgae to survive within a wide range of environmental conditions is largely correlated to the diversity in cellular lipids [50]. Among total lipids, significant differences in fatty acid content were observed between the two different growth conditions (N+ and N−) (Figure 4).

The maximum percentage of saturated fatty acids, monounsaturated, and polyunsaturated (n-3 and n-6) was observed in N+: 19.79 ± 0.78%, 35.06 ± 4.64%, 29.15 ± 3.12%, and 1.83 ± 0.18%, respectively (Figure 4). In N−, there was a significant increase in the amount of saturated and monounsaturated fatty acids: 37.52 ± 1.92% and 49.56 ± 1.74% (Figure 4). The total percentage of polyunsaturated fatty acids was three-fold reduced in N−. The differences were induced by the extensive increase in palmitic (16:0) and palmitoleic acids (16:1n7) and the decrease in eicosapentaenoic acid (20:5n3–EPA), which was abundant in N+ (Table 2) as reported by similar works [14,51,52].

The changes in fatty acids composition caused a significant modification to the polyenoic index (PI) and on the ratio n-3 fatty acids/total fatty acids (n3/tot FA) as shown in Table 3. The polyene index is used to measure the polyunsaturated fatty acids oxidation [53]. EPA and DHA, which represent the main part of polyunsaturated fatty acids (Table 2), were the most susceptible to oxidation. In particular, in N−, EPA and DHA decreased at the increase of 16:0 as a result of the polyene index decreasing significantly. A similar decrease was observed for the total n3/tot FA ratio, caused by a significant decrease in EPA in N−.

Similar results have been reported by Villanova et al. [43] on lipid productivity on the 10th and the 15th days of cultivation of *P. tricornutum* under mixotrophic and phototrophic conditions. *P. tric**ornutum* is a promising candidate for industrial applications for the production of biofuel and of bioactive molecules, but further studies are essential to identify critical control points and a balance between lipid and biomass productivity of microalgae [54].

### 3.3. Cellular Antioxidant Activity

#### 3.3.1. Polyphenol Content

Polyphenols are a very important class of antioxidants that have protective properties associated with cellular defense mechanisms. Although the phenolic content profile varies significantly among species, diatoms adapt differentially to nutritional stress than green algae and plants, reflecting the genetic diversity and complexity of these microorganisms [55]. Significant differences in total polyphenol content between cultivation conditions were observed (Figure 5). Particularly, *P. tricornutum* N+ had a high phenolic compounds content (3.071 ± 0.17 mg GAE/g DW), whereas a lower phenolic content was measured in N− (1.115 ± 0.00 mg GAE/g DW) (Figure 5).

Under standard conditions, redox reactions of reactive oxygen species (ROS) are produced through photorespiration, and it has been established that oxidative phosphorylation by phenolic compounds do protect cells from damage [55]. ROS are markers of oxidative stress related to lipid accumulation and are frequently observed in microalgae grown under abiotic stresses [56].

In agreement with previous studies [55,56], lower concentrations in total polyphenol content were measured in *P. tricornutum*, *Tetraselmis suecica*, and *Chlorella vulgaris* (Chlorophyta) cultivated under N-limited condition [55]. Similarly, in *Tetradesmus dimorphus* (formerly *Acutodesmus dimorphus*) (Chlorophyta), the total polyphenol content initially increased and then decreased after 3 days of cultivation under nitrogen deprivation [56]. Knowledge regarding the impact of nutrient stress on the phenolic content of microalgae still remains scarce and species specific [41,55,57].

#### 3.3.2. Carotenoids

Our results for *P. tricornutum*, cultivated under standard (N+) and starvation conditions (N−), showed a similar pattern to polyphenols content in the carotenoids content (Figure 6), an important class of bioactive compounds with antioxidant properties.

After 15 days of culture, N−s showed the lowest content of carotenoids (0.35 ± 0.01 mg/g DW) compared to N+ (2.70 ± 0.23 mg/g DW). The lowest content of carotenoids in N− was probably the result of alteration of the photosynthetic system in microalgae grown under conditions of nitrogen deprivation, as a common response of photosynthetic organisms to abiotic stressors [58]. Similar results were observed by Gauthier et al. [59] in *P. tricornutum*, *T. suecica*, and *C. vulgaris* cultures maintained under nitrogen starvation. However, other studies described an increase in carotenoid production in microalgae when cultivated under nutrient starvation [55,60].

#### 3.3.3. Total Antioxidant Assays

The high productivity of *P. tricornutum* antioxidants cultivated under standard condition was confirmed by the results of DPPH assay (Figure 7).

In particular, the lowest IC50, 2.29 ± 0.54 mg DW/mL, was recorded in N+. The lowest IC50 corresponded to the highest antioxidant ability. The highest IC50, 3.83 ± 0.66 mg DW/mL, was measured in N−. Considering the pattern observed for polyphenols and carotenoids, these classes of antioxidants are the main contributors to cellular antioxidant activity.

Similar results were obtained by Jeyakumar et al. [61], who observed a higher DPPH scavenging activity in nitrogen-depleted conditions (85%) compared to nitrogen-replete conditions (75%) and control (64%) in *Isochrysis* sp. A study conducted by Singh et al. [62] on *Dunaliella salina* (Chlorophyta) showed similar trends with higher DPPH radical scavenging activity in nitrogen-depleted medium cultures than normal condition. As expected, *P. tricornutum* culture maintained under nitrogen deprivation condition (N−) showed a higher scavenging activity [61].

A further damage on the total antioxidant ability of *P. tric**ornutum* in N− was demonstrated by a reducing power assay. The results shown in Figure 8 are in line with the DPPH assay results. The highest reducing power was observed in *P. tric**ornutum* N+ (EC50, 66.54 ± 2.9 mg DW/mL), whereas in N−, the reducing power decreased (EC50, 147.84 ± 9.85 mg DW/mL).

The obtained results are in accordance with Singh et al. [62], who observed a higher reducing power in nitrogen stressed cells of *Dunaliella salina* than those cultivated under normal condition.

### 3.4. Gene Expression

Diatoms show remarkable diversity among species and a great capacity to adapt their metabolism to different nutritional strategies [56,63]. Relevant genes related to lipid biosynthesis (i.e., *G6PDH*, *ACCa*se, *Cit syn*, and *isocit DH*) and photosynthetic activity (i.e., *rbcL* and *FCP B*) were analyzed in *P. tricornutum* maintained under two different culture conditions (i.e., N+ and N−) (Figure 9).

G6PDH catalyzes the primary reaction of the pentose phosphate pathway (PPP) and produces a large amount of reducing equivalents in the form of nicotinamide adenine dinucleotide phosphate (NADPH), which are essential for lipid biosynthesis [9]. In our experiment, the G6PDH expression was significantly downregulated in *P. tricornutum* N− compared to N+ (*p* < 0.05) (Figure 9), suggesting that the pathway of PPP was negatively affected by nutrient limitation. These results are in contrast with the literature, where overexpression of G6PDH induced an increase in lipid content in *P. tricornutum,* highlighting its critical role in algal lipid accumulation by enhancing NADPH supply [64]. However, transcription of G6PDH in nitrogen limitation shows significant differences between green microalgae *C. vulgaris* [65], *Chlamydomonas reinhardtii* [66], diatom *Thalassiosira pseudonana* [67], and *Nannochloropsis gaditana* (Eustigmatophyceae) [68]. In fact, modifications at the transcript level of the gene G6PDH suggest that PPP is less active in N− conditions, indicating that NADPH yield could be produced by another metabolic pattern. Further investigations are needed to deeply understand the molecular mechanism of G6PDH under starvation conditions.

De novo fatty acid synthesis is one of the important contributors to the total fatty acid pool in the cell [50]. ACCase catalyzes the first step for the de novo fatty acid biosynthesis, and the transcriptional pattern of this gene is very crucial for lipid productivity [69,70]. Acetyl-CoA has a significant role in the central metabolic flux, as it is a primary intermediate for biomolecule production and energy metabolism. ACCase catalyzes the conversion of acetyl-CoA to malonyl-CoA, the substrate for the first step of elongation of fatty acids through FAS (fatty acid synthetase). Transcriptomic and proteomic studies suggested that under nitrogen starvation conditions, plastidial acetyl-CoA carboxylase (one of the two isoforms in microalgae) [22,50] is upregulated, and it represents the key enzyme for de novo fatty acid synthesis pathway [50]. 

In addition, various transcriptomics studies in diatom have revealed the vital role of lipid recycling (as opposed to de novo lipid synthesis) in increasing triacylglycerol accumulation during nitrogen deprivation [22]. Our results showed similar responses in both cultivation conditions (Figure 9). We assumed that the lipid content might influence the activity of the enzyme by a retroactive mechanism depending on the amount of lipid in the cell. These results are in agreement with Guerra et al. [37], indicating that in starvation conditions, the recycling of existing fatty acids occurs rather than de novo fatty acid biosynthesis.

The analysis of gene expression of two important enzymes participating in the tricarboxylic acid cycle (TCA cycle), *Cit syn* and *isocit DH*, was evaluated. Citrate synthase catalyzes the condensation of Acetyl-CoA and oxaloacetate into citrate [71]. Isocitrate dehydrogenase is involved in a critical irreversible reaction of the TCA cycle associated with carbon and nitrogen metabolism, catalyzing the reaction for the formation of α-ketoglutarate, the precursor for amino acid biosynthesis and NADPH [15,50]. The results showed a downregulation of the *Cit syn* gene in N− compared to N+ (Figure 9), suggesting that carbon was redirected from the TCA cycle to lipid metabolism [72]. Our results are in agreement with studies conducted on the green algae *Chlamydomonas reinhardtii*, where mRNA citrate synthase level under starvation conditions were undetectable, highlighting the key role of this gene in lipid biosynthesis, further confirming the relationship between citrate synthase and lipid accumulation [73]. *Isocit DH* was significantly upregulated in N− (*p* < 0.05) (Figure 9). Our results, in accordance with other studies, indicate the importance of these two regulatory enzymes for metabolic flux checkpoints [50,74,75].

The expression of genes involved in photosynthetic carbon fixation (i.e., *rbcL* and *FCP B*) was investigated. The *rbcL* gene encodes the catalytic large subunit of the enzyme rubisco that catalyzes carbon fixation in the first Calvin cycle’s reaction [76]. Carbon dioxide is the exclusive source of carbon in photoautotrophic organisms and is the only pathway for lipid synthesis [76]. *FCP B* is a unique light-harvesting system in diatoms that is scarcely understood and associated with photosystems II. Fucoxanthin is the most abundant carotenoid in *P. tricornutum*, and it is the most crucial pigment in this pattern [77]. Transcript levels of these vital enzymes showed similar variations, as expected. Both genes were severely downregulated in N− (*p* < 0.05) (Figure 9). The intense downregulation of *FCP B* is in accordance with the decrease in carotenoid content in N− as described in Figure 5. Nitrogen limitation in *P. tricornutum* can lead to high damage of the photosynthetic apparatus and modification of enzymes associated with NADPH, the TCA cycle, and PPP [47,78]. This result indicates that nitrogen starvation decelerates the light-harvesting process in photosynthesis and that the photosynthetical activity inhibits the relevant carbon fixation pathway in favor of lipid biosynthesis [21]. Therefore, alternative metabolic pathways may occur, compensating for carbon assimilation, leading to high lipid accumulation under nitrogen starvation [21].

Even though there have been many attempts dedicated to the quantification of *P. tricornutum*’s lipid content [79], transcriptome/proteome/metabolome analyses [75], and to the optimization of biomass production in nitrogen-limited conditions [14], there is still a considerable amount of ambiguity with regard to the molecular mechanisms and modifications of nutrient limitation in this marine diatom. These genes could be used as indicators for environmental and industrial monitoring of *P. tricornutum*.

## 4. Conclusions

Although lipid productivity is a common strategy for microalgae species to survive under abiotic stress condition, cellular mechanisms involved are not fully understood. Our study focused on the molecular changes between lipid biosynthesis and photosynthesis under nitrogen deprivation conditions, associated with the transcriptional expression pattern of central metabolic regulatory enzymes. We observed substantial differences between lipid content, antioxidant productivity, and gene expression patterns. We reported a reduction in EPA and antioxidant productivity and an increase in TAGs in N− cells. These changes are very important from a biotechnological perspective. The changes in the productivity of important secondary metabolites, such as EPA, antioxidants, and TAGs, make *P. tric**ornutum* a potential candidate for pilot-scale cultivation in both nutritional strategies. The gene expression modification of the enzymes studied in this report can be used as markers for environmental and industrial monitoring of *P. tricornutum*.

## Figures and Tables

**Figure 1 antioxidants-11-00411-f001:**
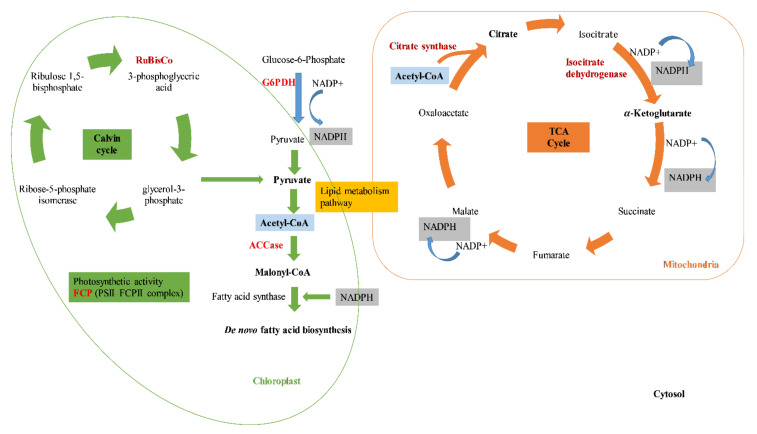
Representative diagram of the enzymes (in red) related to lipid biosynthesis and photosynthesis pathways in *Phaeodactylum tricornutum*. Cytosol: glucose-6-phosphate dehydrogenase (*G6PDH*); chloroplast: acetyl-coenzyme A carboxylase (*ACCase*); citrate synthase (*Cit syn*); isocitrate dehydrogenase (*isocit DH*); mitochondria: ribulose-1,5-bisphospate carboxylase/oxygenase (*RBCL*); fucoxanthin-chlorophyll a/c binding protein B (*FCP B*).

**Figure 2 antioxidants-11-00411-f002:**
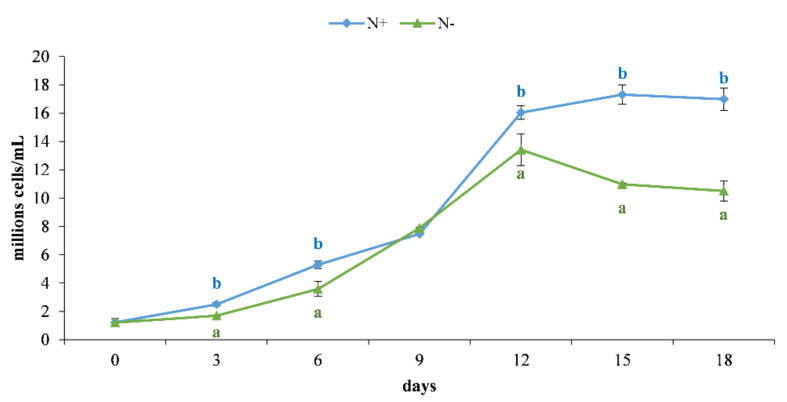
Growth performance of *P. tricornutum* cultivated for 18 days under standard (N+, blue line) and starvation condition (N−, green line). Values are presented as the mean ± SEM (*n* = 3) with standard deviations. Different letters within the same day indicate significant differences (*p* < 0.05) within the two treatment conditions (i.e., N+ and N−).

**Figure 3 antioxidants-11-00411-f003:**
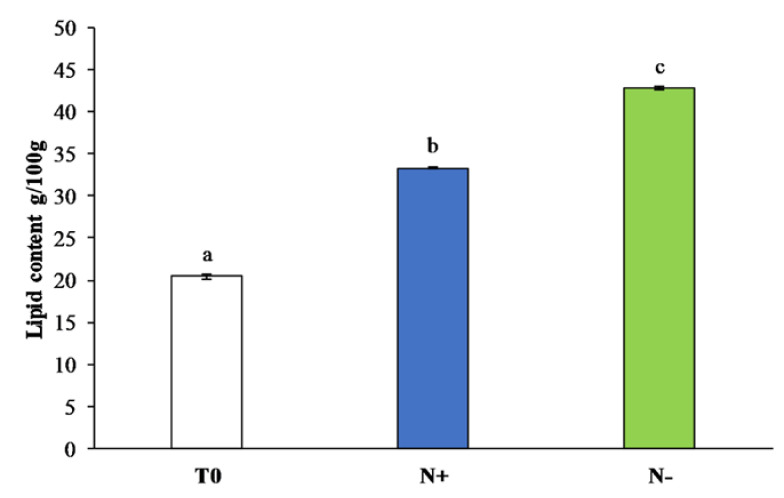
Total lipids (g/100 g) of *P. tricornutum* cultivated for 15 days under initial culture conditions (T0, white bar), under standard conditions (N+, blue bar), and under starvation conditions (N−, green bar). Values are expressed as the mean ± SEM (*n* = 3) with standard deviations. Different letters indicate statistical differences (*p* < 0.05) between groups.

**Figure 4 antioxidants-11-00411-f004:**
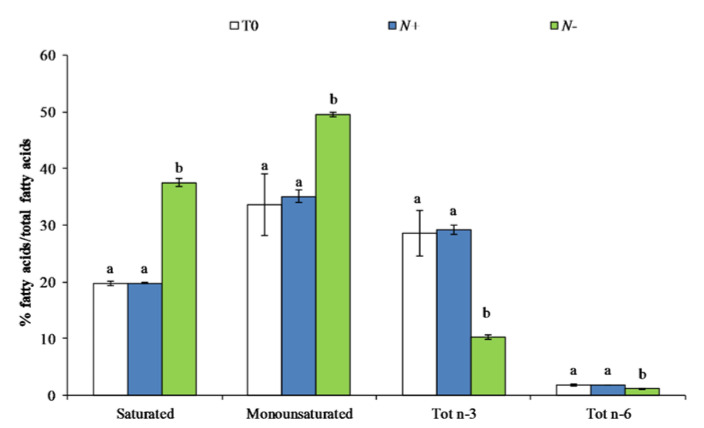
Fatty acids class composition (%, *w/w*) of *P. tricornutum* cultivated for 15 days under initial culture condition (T0, white bar), under standard conditions (N+, blue bar), and under starvation conditions (N−, green bar). Values are expressed as the mean ± SEM (*n* = 3) with standard deviations. Different letters indicate statistical differences (*p* < 0.05) between groups.

**Figure 5 antioxidants-11-00411-f005:**
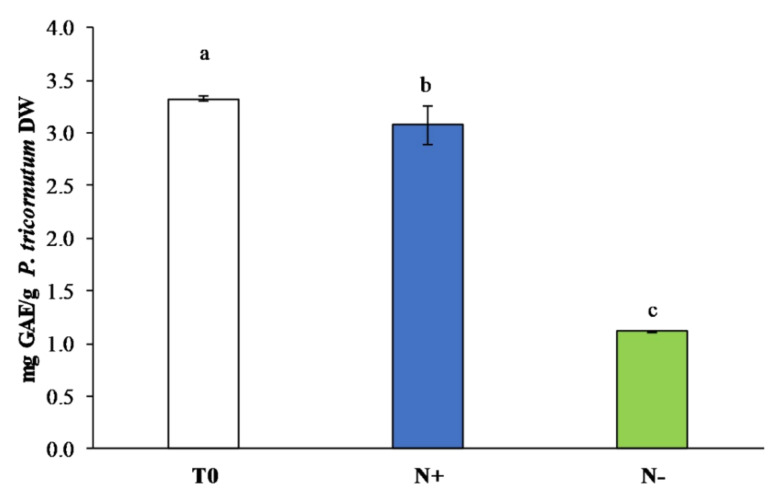
Total polyphenol content (mg GAE/g DW) in *P. tricornutum* extracts cultivated for 15 days under initial culture conditions (T0, white bar), under standard conditions (N+, blue bar), and under starvation conditions (N−, green bar). Values are reported as the means (*n* = 3), and error bars report the standard deviations. Treatments that do not share the same letter were significantly different from each other (*p* < 0.05).

**Figure 6 antioxidants-11-00411-f006:**
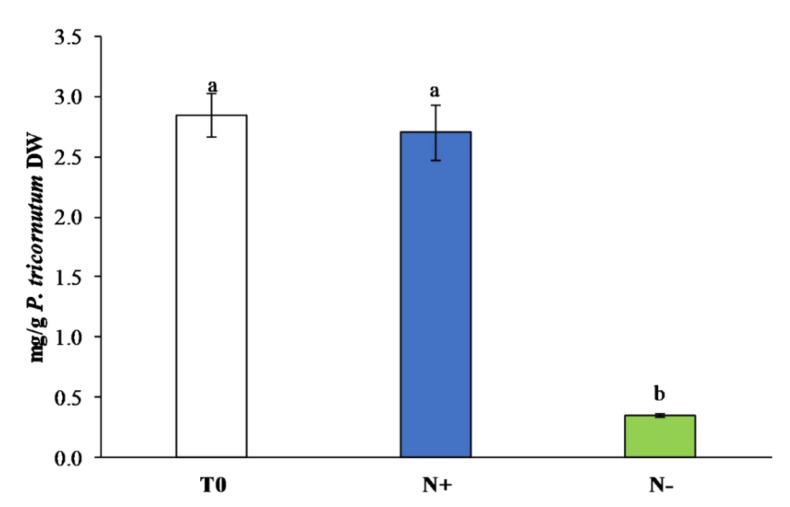
Carotenoid content (mg/g DW) in *P. tricornutum* extracts cultivated for 15 days under initial culture conditions (T0, white bar), under standard conditions (N+, blue bar), and under starvation conditions (N−, green bar). Values are reported as the means (*n* = 3), and error bars report the standard deviations. Treatments that do not share the same letter were significantly different from each other (*p* < 0.05).

**Figure 7 antioxidants-11-00411-f007:**
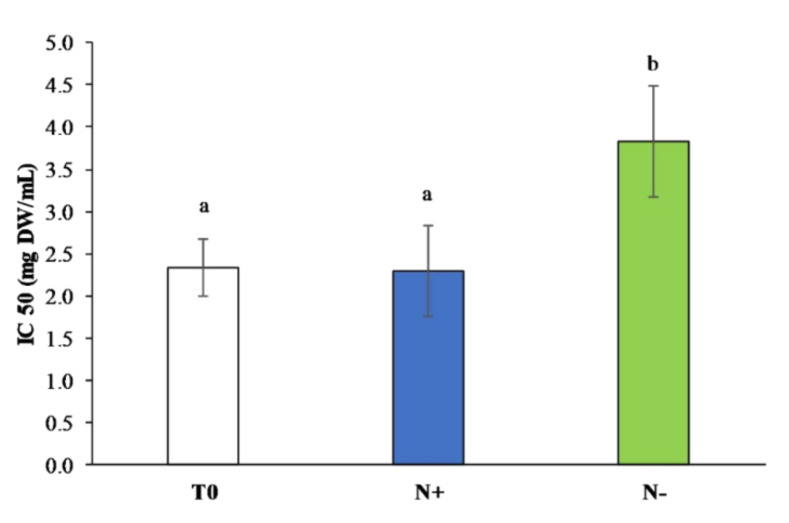
DPPH radical scavenging activity (IC 50, mg DW/mL) in *P. tricornutum* extracts cultivated for 15 days under initial culture conditions (T0, white bar), under standard conditions (N+, blue bar), and under starvation conditions (N−, green bar). Values are reported as the means (*n* = 3), and error bars report the standard deviations. Treatments that do not share the same letter were significantly different from each other (*p* < 0.05).

**Figure 8 antioxidants-11-00411-f008:**
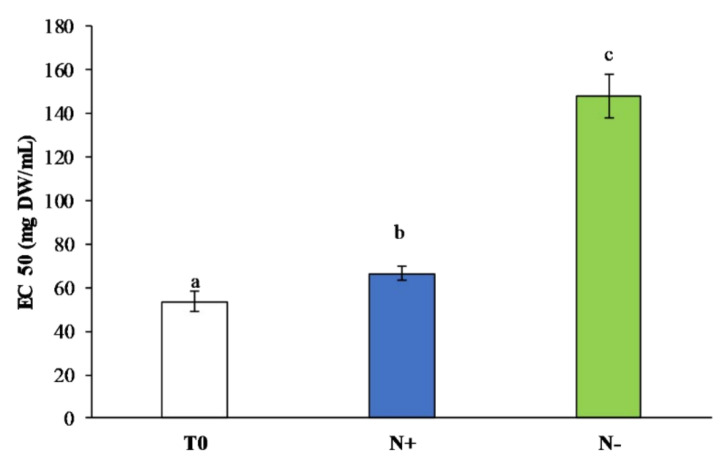
Reducing power (EC50, mg DW/mL) in *P. tricornutum* extracts cultivated for 15 days under initial culture conditions (T0, white bar), under standard conditions (N+, blue bar), and under starvation conditions (N−, green bar). Values are reported as the means (*n* = 3), and error bars report the standard deviations. Treatments that do not share the same letter were significantly different from each other (*p* < 0.05).

**Figure 9 antioxidants-11-00411-f009:**
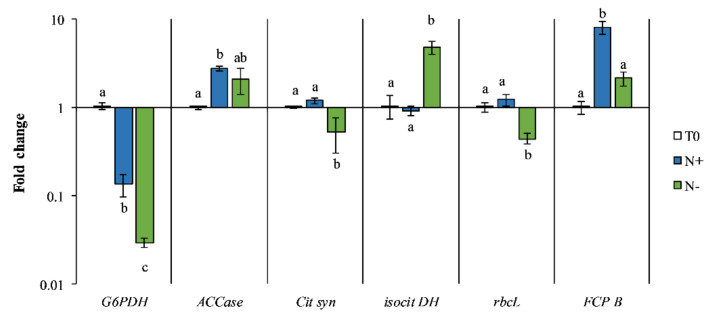
Relative gene expression of genes related to lipid biosynthesis (glucose-6-phosphate dehydrogenase (*G6PDH*), acetyl-coenzyme A carboxylase (*ACCase*), citrate synthase (*Cit syn*), and isocitrate dehydrogenase (*isocit DH*)) and photosynthetic activity (ribulose-1,5-bisphospate carboxylase/oxygenase (large subunit) (*rbcL*) and fucoxanthin-chlorophyll a/c binding protein B (*FCP B*)) in *P. tricornutum* under initial culture conditions (T0, white bars) and after 15 days of cultivation under standard conditions (N+, blue bars) and starvation conditions (N−, green bars). Values are the mean ± SEM (*n* = 3) with standard deviations. Statistical differences (*p* < 0.05) between groups are indicated by different letters.

**Table 1 antioxidants-11-00411-t001:** *Phaeodactylum tricornutum* primer sequences used for real-time PCR.

Gene	Access Number	F/R Primer Sequence (5′–3′)	References
G6PDH		F. GCGAGAAATGGCACAAGG R. GTTCATCGCAGTCGGGAGA	[35]
rbcL	MH064127.1	F. CCAAGGTCCTGCTACTGGTG R. TCTCCAACGCATGAAGGGT	
FCP B		F. GCCGATATCCCCAATGGATTT R. CTTGGTCGAAGGAGTCCCATC	[36]
ACCase,		F. GTTGCTTGACGCTGAACTGG R. CCTTCATGCGACCTGTCTTG	[37]
Cit syn		F. TTATGAAGTCATGCCCGACA R. GGTCCCAGTACAGTTGCGAT	[37]
Isocit DH		F. GGGCAGTCATGAAAGACGTT R. ATCCGTCAGCATATCACCGT	[37]
RPS		F. AATTCCTCGAAGTCAACCAGG R. GTGCAAGAGACCGGACATAC	[38]
TBP		F. ATCGATTTGTCAATCCACGAG R. ATACAGATTCTGTGTCCACGG	[38]

**Table 2 antioxidants-11-00411-t002:** Fatty acids composition (%, *w/w*) of *P. tricornutum* cultivated for 15 days under standard (N+) and starvation condition (N−).

Fatty Acid	N+	N−
14:0	7.35 ± 0.54 ^a^	4.85 ± 0.61 ^b^
16:0	11.87 ± 0.72 ^a^	31.90 ± 1.57 ^b^
16:1n-7	31.12 ± 2.82 ^a^	43.35 ± 1.59 ^b^
16:2n-4	1.99 ± 0.62 ^a^	0.38 ± 0.02 ^b^
16:3n-4	12.12 ± 1.51 ^a^	1.07 ± 0.10 ^b^
18:1n-9	2.63 ± 1.59	4.90 ± 0.34
18:1n-7	1.13 ± 0.21	1.16 ± 0.18
18:2n-6	1.11 ± 0.20 ^a^	0.71 ± 0.11 ^b^
20:5n-3 EPA	24.43 ± 1.85 ^a^	8.30 ± 0.80 ^b^
22:5n-3	1.88 ± 0.37 ^a^	0.69 ± 0.13 ^b^
22:6n-3 DHA	1.62 ± 0.33 ^a^	0.71 ± 0.10 ^b^

Data are reported as the means (*n* = 3) with standard deviation. Different letters indicate significant differences between each treatment (*p* < 0.05).

**Table 3 antioxidants-11-00411-t003:** Unsaturation markers calculated from the fatty acid profile of *P. tricornutum* cultivated for 15 days under standard (N+) and starvation condition (N−).

	N+	N−
Polyene Index	2.21 ± 0.32 ^a^	0.28 ± 0.04 ^b^
n3/tot FA	0.30 ± 0.04 ^a^	0.10 ± 0.01 ^b^

Data are reported as the means (*n* = 3) with standard deviation. Different letters indicate significant differences between each treatment (*p* < 0.05).

## Data Availability

Data contained within the article.

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
