# Peer review of "Culture Conditions Affect Antioxidant Production, Metabolism and Related Biomarkers of the Microalgae Phaeodactylum tricornutum"

_antioxidants, 2022, doi:10.3390/antiox11020411_

Round 1
Reviewer 1 Report
The authors submitted an interesting original article, which deals with effects of cultivation conditions on metabolism and the antioxidant production of the algal species Phaeodactylum tricornutum (Bacillariophyceae). Since various algal species of class Bacillariophyceae are cosmopolite, occurring in marine and fresh water as well as in soil, the topic of this manuscript is important for the scientific community, especially phycologists. Additionally, it is assumed that various algae could become a possible main source of food for human population in the future.
The authors cultivated the species in various cultivation media, especially with different source of nitrogen for the algal cells. Consequently, the content of carotenoids, lipids (included saturated and unsaturated fatty acids), antioxidant activity and other parameters were explored. The authors used adequate methods and the experiments were well-planned. The results are properly discussed and well-presented by several illustrative figures and tables. Furthermore, the results were correctly interpreted and they supported author’s conclusions. Additionally, in the manuscript, the authors cited the most relevant references.
Minor issue:
I recommend increasing resolution of Figure 1 because the text in this figure can be read with difficulties.
Author Response
Referee 1
The authors submitted an interesting original article, which deals with effects of cultivation conditions on metabolism and the antioxidant production of the algal species Phaeodactylum tricornutum (Bacillariophyceae). Since various algal species of class Bacillariophyceae are cosmopolite, occurring in marine and fresh water as well as in soil, the topic of this manuscript is important for the scientific community, especially phycologists. Additionally, it is assumed that various algae could become a possible main source of food for human population in the future.
The authors cultivated the species in various cultivation media, especially with different source of nitrogen for the algal cells. Consequently, the content of carotenoids, lipids (included saturated and unsaturated fatty acids), antioxidant activity and other parameters were explored. The authors used adequate methods and the experiments were well-planned. The results are properly discussed and well-presented by several illustrative figures and tables. Furthermore, the results were correctly interpreted and they supported author’s conclusions. Additionally, in the manuscript, the authors cited the most relevant references.
Minor issue:
I recommend increasing resolution of Figure 1 because the text in this figure can be read with difficulties.
Answer: The authors wish to thank the Reviewer 1 for the comments. We revised the manuscript by improving resolution of Figure 1, as kindly suggested.
Reviewer 2 Report
The manuscript "Culture Conditions Affect Antioxidant Production, Metabolism and Related Biomarkers of the Microalgae Phaeodactylum Tricornutum" addresses a relevant and appropriate topic for this journal. Authors should make corrections to the algae taxonomy
Corrections needed:
line 3/4 - of the Microalgae Phaeodactylum tricornutum
line 26 - Abstract: Phaeodactylum tricornutum (Bacillariophyta) is a worldwide-distributed diatom ...
line 116 - ... for 20 min (Eppendorf Centrifuge 5430 R).
line 151 ... Absorbance was read after 30 min
line 207 - Chlorella vulgaris (Chlorophyta) [40], ...
line 223/224 - on Nannochloropsis sp. (Ochrophyta, Eustigmatophyceae) [44], Conticribra weissflogii (formerly Thalassiosira weissflogii) and Cyclotella cryptica (Bacillariophyta) [45] maintained under...
line 293 - ... Tetraselmis suecica and Chlorella vulgaris (Chlorophyta) ...
line 294 ... Tetradesmus dimorphus (formerly Acutodesmus dimorphus) (Chlorophyta) the total ...
line 333 - ... Dunaliella salina (Chlorophyta) ...
line 376 - ... diatom Thalassiosira ...
line 377 - ... and Nannochloropsis gaditana (Eustigmatophyceae) [64].
lines 468 to 636 - The scientific names in the bibliographic references must all be in italics
Author Response
Referee 2
The manuscript "Culture Conditions Affect Antioxidant Production, Metabolism and Related Biomarkers of the Microalgae Phaeodactylum Tricornutum" addresses a relevant and appropriate topic for this journal. Authors should make corrections to the algae taxonomy.
Corrections needed:
line 3/4 - of the Microalgae Phaeodactylum tricornutum
line 26 - Abstract: Phaeodactylum tricornutum (Bacillariophyta) is a worldwide-distributed diatom..
line 116 - ... for 20 min (Eppendorf Centrifuge 5430 R).
line 151 ... Absorbance was read after 30 min
line 207 - Chlorella vulgaris (Chlorophyta) [40], ...
line 223/224 – on Nannochloropsis sp. (Ochrophyta, Eustigmatophyceae) [44], Conticribra weissflogii (formerly Thalassiosira weissflogii) and Cyclotella cryptica (Bacillariophyta) [45] maintained under...
line 293 - ...Tetraselmis suecica and Chlorella vulgaris (Chlorophyta) ...
line 294 ...Tetradesmus dimorphus (formerly Acutodesmus dimorphus) (Chlorophyta) the total ...
line 333 - ... Dunaliella salina (Chlorophyta) ...
line 376 - ... diatom Thalassiosira...
line 377 - ... and Nannochloropsis gaditana (Eustigmatophyceae) [64].
lines 468 to 636 - The scientific names in the bibliographic references must all be in italics
Answer: The authors wish to thank the Reviewer 2 for the comments. We revised the manuscript according to the valuable comments that helped to improve the paper.